# Mesolimbic dopamine ramps reflect environmental timescales

Joseph R Floeder[1], Huijeong Jeong[2], Ali Mohebi[2,3,4], Vijay Mohan K Namboodiri[1,2,5]*

[1]Neuroscience Graduate Program, University of California, San Francisco, San Francisco, United States; [2]Department of Neurology, University of California, San Francisco, San Francisco, United States; [3]Department of Psychology, University of Wisconsin–Madison, Madison, United States; [4]Neuroscience Training Program, University of Wisconsin–Madison, Madison, United States; [5]Weill Institute for Neurosciences, Kavli Institute for Fundamental Neuroscience, Center for Integrative Neuroscience, University of California, San Francisco, San Francisco, United States

*For correspondence:
VijayMohan.KNamboodiri@ucsf.edu

Competing interest: The authors declare that no competing interests exist.

## eLife Assessment

Floeder and colleagues provide an **important** investigation that describes the experimental conditions that systematically produce "ramps" in dopamine signaling in the striatum. This somewhat nebulous feature of dopamine has been a significant part of recent theoretical and computational debates attempting to formally describe the different timescales on which dopamine functions. The current results are **convincing** and add context to that ongoing work.

**Abstract** Mesolimbic dopamine activity occasionally exhibits ramping dynamics, reigniting debate on theories of dopamine signaling. This debate is ongoing partly because the experimental conditions under which dopamine ramps emerge remain poorly understood. Here, we show that during Pavlovian and instrumental conditioning in mice, mesolimbic dopamine ramps are only observed when the inter-trial interval is short relative to the trial period. These results constrain theories of dopamine signaling and identify a critical variable determining the emergence of dopamine ramps.

## Introduction

Mesolimbic dopamine activity was classically thought to operate in either a 'phasic' or a 'tonic' mode (*Berke, 2018*; *Grace, 1991*; *Niv et al., 2007*). Yet, recent evidence points to a 'quasi-phasic' mode in which mesolimbic dopamine activity exhibits ramping dynamics (*Howe et al., 2013*; *Hamid et al., 2016*; *Mohebi et al., 2019*; *Collins et al., 2016*; *Syed et al., 2016*; *Engelhard et al., 2019*; *Guru et al., 2020*; *Kim et al., 2020*; *Hamid et al., 2021*; *Hamilos et al., 2021*; *Gao et al., 2021*; *Farrell et al., 2022*; *Krausz et al., 2023*). This discovery reignited debate on theories of dopamine function because it appeared inconsistent with the dominant theory that dopamine signaling conveys temporal difference reward prediction error (RPE) *Niv, 2013*, since ramping dopamine would paradoxically be a 'predictable prediction error' (*Lloyd and Dayan, 2015*). Recent work has hypothesized that dopamine ramps reflect the value of ongoing states, serving as a motivational signal (*Berke, 2018*; *Howe et al., 2013*; *Hamid et al., 2016*; *Mohebi et al., 2019*). Others have argued that ramping dopamine indeed reflects RPE under some assumptions, namely correction of uncertainty via sensory feedback (*Kim et al., 2020*; *Farrell et al., 2022*; *Mikhael et al., 2022*), representational error (*Gershman, 2014*), or memory lapses (*Morita and Kato, 2014*). Still others have proposed that dopamine ramps reflect

a causal influence of actions on rewards in instrumental tasks (*Hamid et al., 2021*). This debate has been exacerbated in part because there is no clear understanding of why dopamine ramps appear only under some experimental conditions. Accordingly, uncovering a unifying principle of the conditions under which dopamine ramps appear will provide important constraints on theories of dopamine function (*Berke, 2018*; *Hamid et al., 2021*; *Lloyd and Dayan, 2015*; *Mikhael et al., 2022*; *Gershman, 2014*; *Morita and Kato, 2014*; *Redgrave and Gurney, 2006*; *Berridge, 2007*; *Bromberg-Martin et al., 2010*; *Waddell, 2013*; *Gardner et al., 2018*; *Saunders et al., 2018*; *Sharpe et al., 2020*; *Hughes et al., 2020*; *Maes et al., 2020*; *Kutlu et al., 2021*; *Jeong et al., 2022*; *Jakob et al., 2022*; *Coddington et al., 2023*; *Masset et al., 2025*; *Sousa et al., 2025*; *Takahashi et al., 2023*; *Markowitz et al., 2023*; *Carter et al., 2024*; *Tang et al., 2024*; *Sias et al., 2024*; *Garr et al., 2024*).

To investigate the necessary conditions for dopamine ramps, we turned to our recent work proposing that dopamine acts as a teaching signal for causal learning by representing the Adjusted Net Contingency for Causal Relations (ANCCR; *Jeong et al., 2022*; *Garr et al., 2024*; *Burke et al., 2023*). The crux of the ANCCR model is that learning to predict a future meaningful event (e.g. reward) can occur by looking back in time for potential causes of that event. Critically, the ability to learn by looking backwards depends on how long one holds on to past events in memory. If the past is quickly forgotten, there is little ability to identify causes that occurred long before a reward. On the other hand, maintaining memory for too long is computationally inefficient and would allow illusory associations across long delays. Thus, for optimal learning, the timescale for memory maintenance should flexibly depend on 'environmental timescales' set by the overall rates of events. In ANCCR, this is achieved by controlling the duration of a memory trace of past events with the 'eligibility trace' time constant (illustrated in *Figure 1—figure supplement 1*). When there is a high average rate of events, the eligibility trace time constant is small. Accordingly, we successfully simulated dopamine ramping dynamics assuming two conditions: a dynamic progression of cues that signal temporal proximity to reward, and a small eligibility trace time constant relative to the trial period (*Jeong et al., 2022*). However, whether these conditions are sufficient to experimentally produce mesolimbic dopamine ramps in vivo remains untested.

In this study, we designed experiments to address the influence of environmental timescales on dopamine ramps. Specifically, we sought to test the key prediction that dopamine ramps would be observed for a small, but not large, eligibility trace time constant, which is hypothesized to emerge for high overall event rate. To do so, we manipulated the inter-trial interval (ITI) duration in both an auditory Pavlovian conditioning paradigm and a virtual reality navigation task. The results confirmed our key prediction from ANCCR, providing a clear constraint on theoretical explanations for the controversial phenomenon of dopamine ramps.

## Results

We first measured mesolimbic dopamine release in the nucleus accumbens core using a dopamine sensor (dLight1.3b; *Patriarchi et al., 2018*) in an auditory cue-reward task. We varied both the presence or absence of a progression of cues indicating reward proximity ('dynamic' vs 'fixed' tone) and the inter-trial interval (ITI) duration (short vs long ITI). Varying the ITI was critical because our theory predicts that the ITI is a variable controlling the eligibility trace time constant, such that a short ITI would produce a small time constant relative to the cue-reward interval (**Appendix 1**, *Figure 1A-E*). In all four experimental conditions, head-fixed mice learned to anticipate the sucrose reward, as reflected by anticipatory licking (*Figure 1F–G*). In line with our earlier work, we showed that simulations of ANCCR exhibit a larger cue onset response when the ITI is long and exhibit ramps only when the ITI is short (*Figure 1H*). Consistent with these simulated predictions, experimentally measured mesolimbic dopamine release had a much higher cue onset response for long ITI (*Figure 1I–J*). Furthermore, dopamine ramps were observed only when the ITI was short and the tone was dynamic (*Figure 1I and K–M*, *Figure 1—figure supplement 2*). Indeed, dopamine ramps—quantified by a positive slope of dopamine response vs time within trial over the last five seconds of the cue—appeared on the first day after transition from a long ITI/dynamic tone condition to a short ITI/dynamic tone condition and disappeared on the first day after transition from a short ITI/dynamic tone condition to a short ITI/fixed tone condition (*Figure 1L*). These results confirm the key prediction of our theory in Pavlovian conditioning.

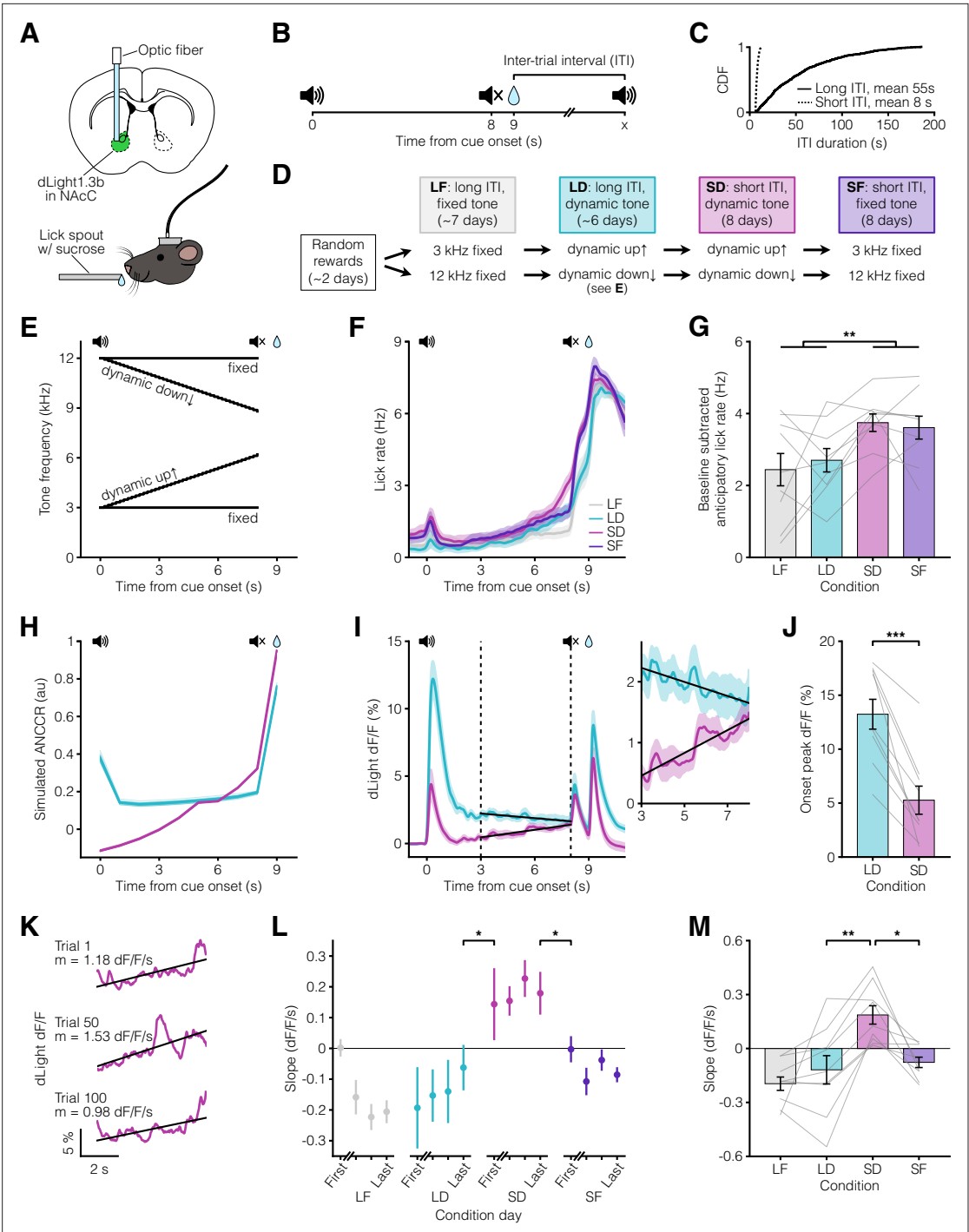

**Figure 1.** Pavlovian conditioning dopamine ramps depend on ITI. (A) Top, fiber photometry approach schematic for nucleus accumbens core (NAcC) dLight recordings. Bottom, head-fixed mouse. (B) Pavlovian conditioning experimental setup. Trials consisted of an 8 s auditory cue followed by sucrose reward delivery 1 s later. (C) Cumulative Distribution Function (CDF) of ITI duration for long (solid line, mean 55 s) and short ITI (dashed line, mean 8 s) conditions. (D) Experimental timeline. Mice were divided into groups receiving either a 3 kHz fixed and dynamic up↑ tone or a 12 kHz fixed and dynamic down↓ tone. (E) Tone frequency over time. (F) Peri-stimulus time histogram (PSTH) showing average licking behaviors for the last 3 days of each condition (n=9 mice). (G) Average anticipatory lick rate (baseline subtracted) for 1 s preceding reward delivery (long ITI vs short ITI: **p=0.0046). (H) ANCCR simulation results from an 8 s dynamic cue followed by reward 1 s later for long ITI (teal) and short ITI (pink) conditions. Bold lines show the average of 20 iterations. (I) Left, average dLight dopamine signals. Vertical dashed lines represent the ramp window from 3 to 8 s after cue onset, thereby excluding the influence of the cue onset and offset responses. Solid black lines show linear regression fit during the window. Right, closeup of dopamine signal during window. (J) Average peak dLight response to cue onset for LD and SD conditions (***p=1.9 x 10⁻⁴). (K) dLight dopamine signal

*Figure 1 continued on next page*

*Figure 1 continued*

with linear regression fit during ramp window for example SD trials. Reported m is slope. (**L**) Session average per-trial slope during ramp window for the first day and last 3 days of each condition (last day LD to first day SD: *p=0.036, last day SD to first day SF: *p=0.023). (**M**) Average per-trial slope for last 3 days of each condition (LD vs SD: **p=0.0026, SD vs SF: *p=0.011). All data presented as mean ± SEM. See ***Supplementary file 1*** for full statistical details.

The online version of this article includes the following figure supplement(s) for figure 1:

**Figure supplement 1.** Dependence of ANCCR on eligibility trace time constant.

**Figure supplement 2.** Pavlovian conditioning cohort 1 histology and dopamine responses.

While these results are consistent with the idea that dopamine ramps are shaped by the ITI, an alternative explanation could be differences in behavioral learning across experimental conditions. To test this possibility, we repeated the same Pavlovian conditioning paradigm with a counterbalanced training order in a second cohort of mice (***Figure 2A***). Despite the shuffled training order, this cohort behaved similarly and showed robust anticipatory licking across conditions (***Figure 2—figure supplement 1A–B***). As with the previous cohort, dopamine ramps were only observed in the short ITI/dynamic tone condition, rapidly appearing on the first day of this condition and disappearing on the first day of the subsequent long ITI/dynamic tone condition (***Figure 2B–D***, ***Figure 2—figure supplement 2***, ***Figure 2—figure supplement 3***). Critically, the presence of dopamine ramps during the last five seconds of the cue could not be explained by variations in behavior; during this period, anticipatory licking was similar across all conditions, and there was no difference in the slope of the lick rate between the dynamic tone conditions (***Figure 2—figure supplement 1C–D***). Taken together, these results rule out any effects of differential learning across conditions on dopamine ramps.

Although the difference in dopamine ramp slope seems to be well explained by the ITI condition, it might instead reflect differences in post-reward dopamine dynamics, which drop below baseline. As dopamine levels recover from this drop over several seconds, it could appear as a dopamine ramp on the subsequent trial given a sufficiently short ITI. The lack of dopamine ramps in the short ITI/fixed tone condition serves as a control for this, however (***Figure 1L–M***). Furthermore, there is no significant difference in the pre-cue dopamine slope between conditions, nor is there a correlation between the pre-cue dopamine slopes and the dopamine ramp slopes during the cue in the short ITI/dynamic

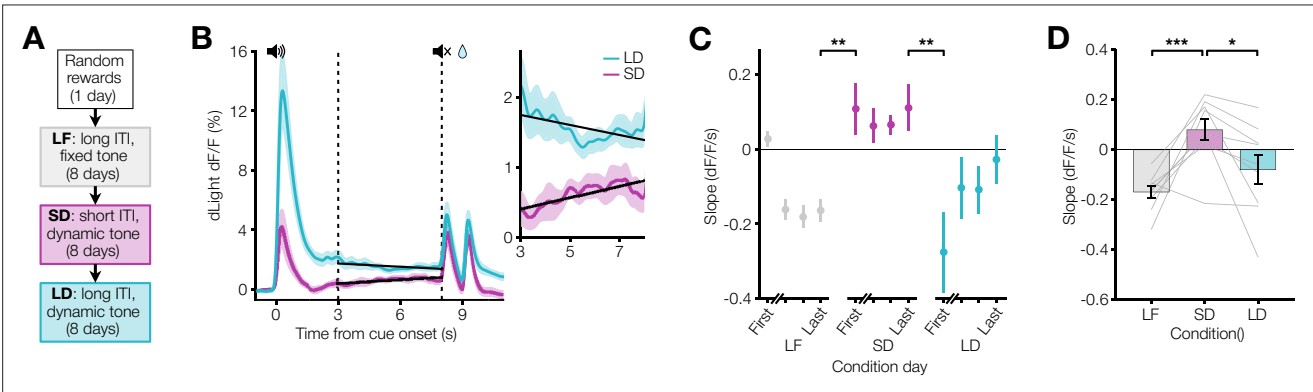

**Figure 2.** Pavlovian conditioning dopamine ramps do not depend on training order. (**A**) Experimental timeline in which the SD condition occurs before the LD condition. (**B**) Left, average dLight dopamine signals for SD and LD conditions. Vertical dashed lines represent the ramp window from 3 to 8 s after cue onset. Solid black lines show linear regression fit during the window. Right, close-up of dopamine signal during window (n=9 mice). (**C**) Session average per-trial slope during ramp window for the first day and last 3 days of each condition (last day LF to first day SD: **p=0.0045, last day SD to first day LD: **p=0.0067). (**D**) Average per-trial slope for last 3 days of each condition (LF vs SD: ***p=9.1 x 10⁻⁴, SD vs LD: *p=0.010).

The online version of this article includes the following figure supplement(s) for figure 2:

**Figure supplement 1.** Pavlovian conditioning licking behavior data.

**Figure supplement 2.** Pavlovian conditioning cohort 2 histology and dopamine responses.

**Figure supplement 3.** Pavlovian conditioning cumulative dopamine data.

**Figure supplement 4.** Pavlovian conditioning dopamine ramps do not correlate with pre-cue dopamine activity.

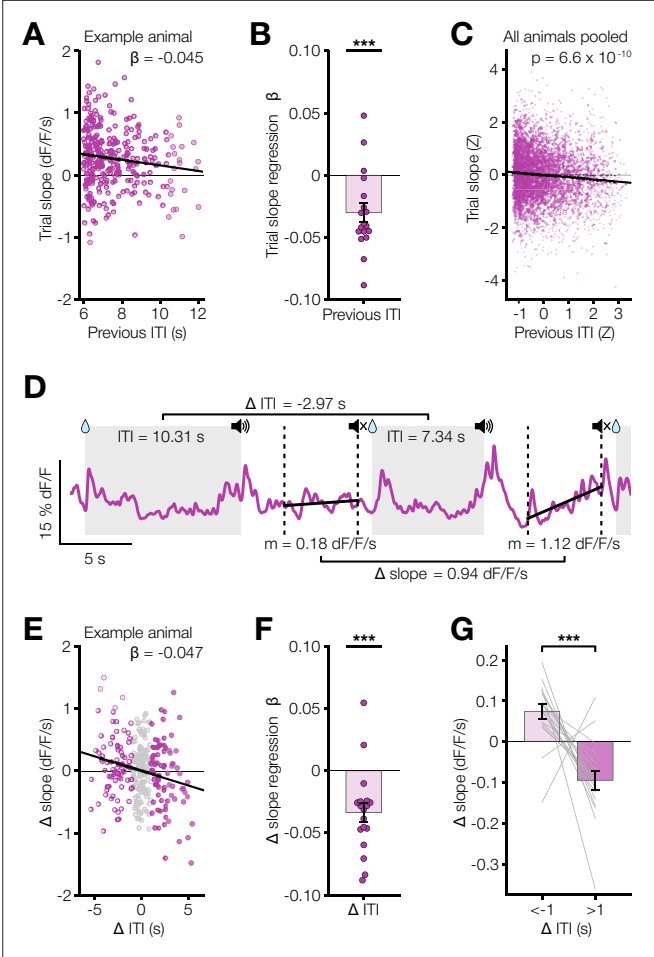

**Figure 3.** Per-trial dopamine ramps correlate with previous ITI. (**A**) Scatter plot for an example animal showing the relationship between dopamine response slope within a trial and previous ITI for all trials in the last 3 days of SD condition. Plotted with linear regression fit (black line) used to find this animal's β coefficient of –0.045. (**B**) Linear regression β coefficients for previous ITI vs. trial slope calculated per animal (***p=5.6 x 10⁻⁴). (**C**) Scatter plot of Z-scored trial slope vs. previous ITI pooled across mice for all trials in the last 3 days of SD condition (***p=6.6 x 10⁻¹⁰). The Z-scoring per animal removes the effect of variable means across animals on the slope of the pooled data. (**D**) dLight dopamine signal for two consecutive example SD trials showing the change in ITI and change in slope. The gray shaded regions indicate ITIs, and the vertical dashed lines mark the ramp window period. Reported m is slope. (**E**) Scatter plot for the same example animal in **A** showing the relationship between the change in dopamine slopes and the change in ITI across all trials in the last 3 days of SD condition. Plotted with linear regression fit (black line). Dot colors indicate magnitude of Δ ITI: light pink for Δ ITI below –1 s; gray for Δ ITI between –1 s and 1 s; dark pink for Δ ITI above 1 s. (**F**) Linear regression β coefficients for Δ ITI vs. Δ slope calculated per animal (***p=3.2 x 10⁻⁴). (**G**) Comparison of the average Δ slope for Δ ITI below –1 s vs above 1 s (*** p=2.3 x 10⁻⁴).

The online version of this article includes the following figure supplement(s) for figure 3:

**Figure supplement 1.** Pavlovian conditioning dopamine responses do not correlate with broader estimates of ITI.

**Figure supplement 2.** No significant correlations exist between additional dopamine and behavior measurements.

condition (*Figure 2—figure supplement 4*). As such, our results cannot be captured by a natural ramp in the dopamine signal following reward.

Given the speed with which dopamine ramps appeared and disappeared, we next tested whether the slope of dopamine ramps in the short ITI/dynamic tone condition depended on the previous ITI duration on a trial-by-trial basis. We found that there was indeed a statistically significant trial-by-trial correlation between the previous ITI duration and the current trial's dopamine response slope in the

short ITI/dynamic condition with ramps, but not in the long ITI/dynamic condition without ramps (*Figure 3A–C*). The dependence of a trial's dopamine response slope with previous ITI was significantly negative, meaning that a longer ITI correlates with a weaker ramp on the next trial. This finding held when analyzing either animal-by-animal (*Figure 3A–B*) or the pooled trials across animals while accounting for mean animal-by-animal variability (*Figure 3C*). This relationship was only significant for a single previous trial, however, and did not hold for a broader estimate of average previous ITIs (*Figure 3—figure supplement 1*). In addition, we quantified how the relative change in ITI duration between consecutive trials correlates with changes in dopamine ramp slope (*Figure 3D*). We found a significantly negative relationship between the change in dopamine slope and change in ITI (*Figure 3E–F*). Furthermore, the change in slope was significantly greater for relative decreases in ITI compared to relative increases in ITI, indicating that a relatively shorter ITI tends to have a stronger ramp (*Figure 3G*). These results suggest that the eligibility trace time constant adapts rapidly to changing ITI in Pavlovian conditioning.

Due to the robust relationship between ITI and dopamine ramp slope on a per-trial basis, we next sought to explore the potential relationships between other important dopaminergic and behavioral variables. Though the dopamine cue onset response is significantly greater in the long compared to short ITI/dynamic tone condition, there is no apparent relationship between the cue onset response and dopamine ramp slope in either condition (*Figure 3—figure supplement 2A–B*). Furthermore, neither the cue onset response nor the dopamine ramp slope correlates with the per-trial behavior quantified as lick slope (*Figure 3—figure supplement 2C–F*). Finally, unlike the ramping dopamine slope, this ramping lick slope did not correlate with ITI duration (*Figure 3—figure supplement 2G–H*). The fact that this exploration of additional variables yielded no significant relationships highlights the unique, specific influence of ITI on dopamine ramp slope.

We next tested whether the results from Pavlovian conditioning could be reproduced in an instrumental task. In keeping with prior demonstrations of dopamine ramps in head-fixed mice, we used a virtual reality (VR) navigational task in which head-fixed mice had to run towards a destination in a virtual hallway to obtain sucrose rewards (*Kim et al., 2020*; *Farrell et al., 2022*; *Mikhael et al., 2022*; *Lopes et al., 2021*; *Figure 4A–B*, *Figure 4—figure supplement 1*). At reward delivery, the screen turned blank during the ITI and remained so until the next trial onset. After training animals in this task using a medium ITI, we changed the ITI duration to short or long for eight days before switching to the other (*Figure 4C*). We found evidence that mice learned the behavioral requirement during the trial period, as they significantly increased their running speed during trial onset (*Figure 4D–E*) and reached a similarly high speed prior to reward in both ITI conditions (*Figure 4F–G*). Consistent with the results from Pavlovian conditioning, the dopamine response to the onset of the hallway presentation was larger during the long ITI compared to the short ITI condition (*Figure 4H–I*), and dopamine ramps were observed only in the short ITI condition (*Figure 4J–M*). Unlike the Pavlovian conditioning, the change in the ITI resulted in a more gradual appearance or disappearance of ramps (*Figure 4L*), but there was still a weak overall correlation between dopamine response slope on a trial and the previous inter-reward interval (*Figure 4—figure supplement 2*). These results are consistent with a more gradual change in the eligibility trace time constant in this instrumental task. Collectively, the core finding from Pavlovian conditioning that mesolimbic dopamine ramps are present only during short ITI conditions was reproduced in the instrumental VR task.

## Discussion

Our results provide a general framework for understanding past results on dopamine ramps. According to ANCCR, the fundamental variable controlling the presence of ramps is the eligibility trace time constant. Based on first principles, this time constant depends on the ITI in common task designs (**Appendix 1**). Thus, the ITI is a simple proxy to manipulate the eligibility trace time constant, thereby modifying dopamine ramps. In previous navigational tasks with dopamine ramps, there was no explicitly programmed ITI (*Howe et al., 2013*; *Guru et al., 2020*; *Krausz et al., 2023*). As such, the controlling of the pace of trials by these highly motivated animals likely resulted in short effective ITI compared to trial duration. An instrumental lever pressing task with dopamine ramps similarly had no explicitly programmed ITI (*Collins et al., 2016*), and other tasks with observed ramps had short ITIs (*Syed et al., 2016*; *Kim et al., 2020*; *Hamilos et al., 2021*; *Gao et al., 2021*; *Farrell et al., 2022*). One reported result that does not fit with a simple control of ramps by ITI is that navigational tasks

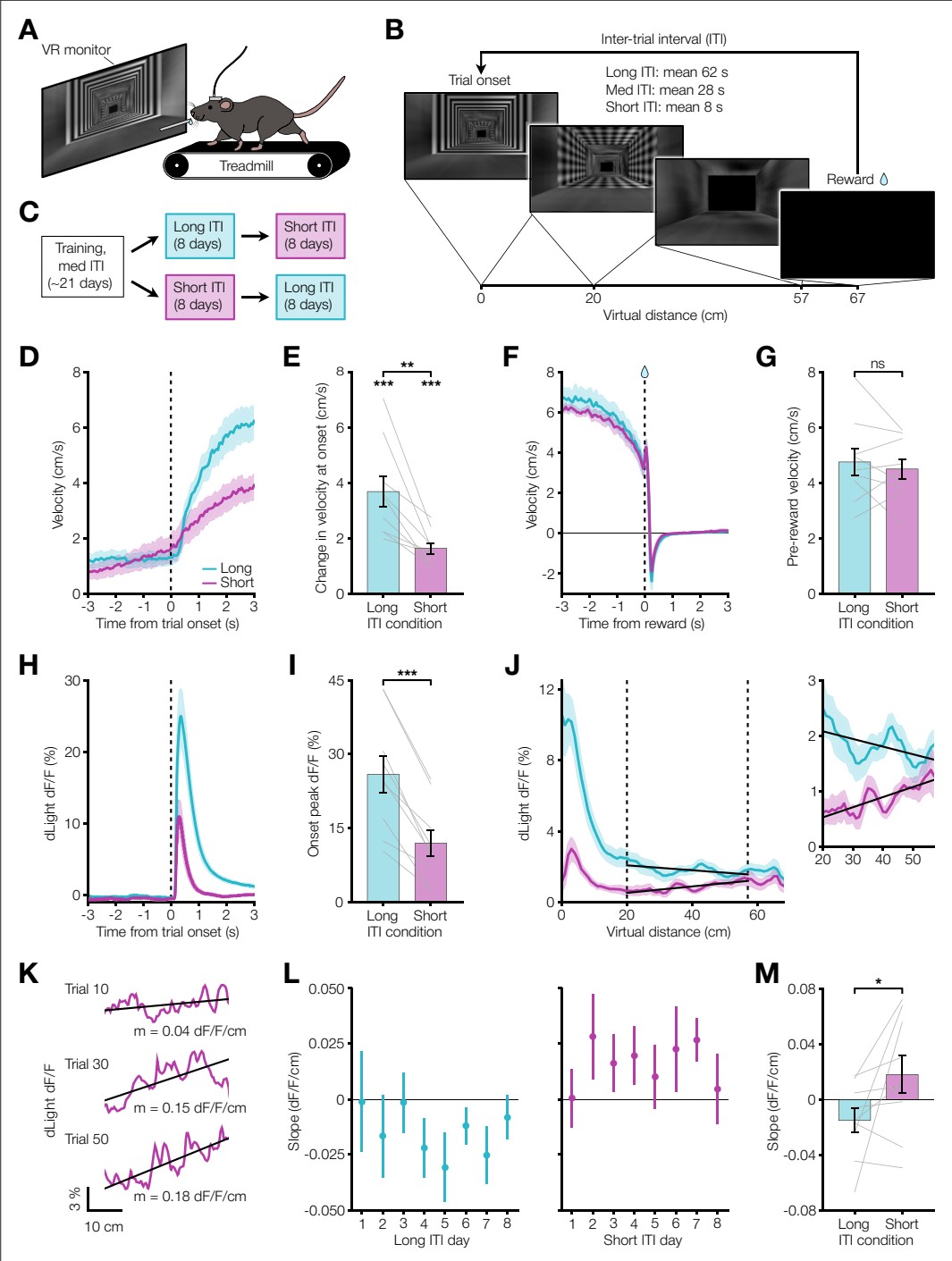

**Figure 4.** VR navigation dopamine ramps depend on ITI. (**A**) Head-fixed VR approach schematic. (**B**) VR navigation task experimental setup. Trials consisted of running down a patterned virtual hallway to receive sucrose reward. VR monitor remained black during the ITI. (**C**) Experimental timeline. Following training, mice were assigned to either long or short ITI conditions for 8 days before switching. (**D**) Velocity PSTH aligned to trial onset for long (teal) and short (pink) ITI conditions (n=9 mice). (**E**) Average change in velocity at trial onset. Bottom asterisks indicate both conditions significantly differ from zero (long: ***p=1.0 x 10⁻⁴, short: ***p=2.3 x 10⁻⁵). Top asterisks indicate significant difference between conditions (**p=0.0028). (**F**) Velocity PSTH aligned to reward delivery. (**G**) Average velocity during 1 s preceding reward (p=0.50). (**H**) PSTH showing average dLight dopamine signal aligned to trial onset. (**I**) Comparison of peak dLight onset response (***p=6.3 x 10⁻⁵). (**J**) Left, average dLight dopamine signal across distances spanning the entire virtual corridor. Vertical dashed lines represent the ramp window from 20 to

*Figure 4 continued on next page*

*Figure 4 continued*

57 cm (10 cm before end of track). Solid black lines show linear regression fit during window. Right, close-up of dopamine signal during window. (**K**) dLight dopamine signal with linear regression fit during ramp window for example short ITI trials. Reported m is slope. (**L**) Session average per-trial slope during ramp window for all days of each condition. (**M**) Comparison of average per-trial slope during ramp window for last 3 days of both conditions (*p=0.035).

The online version of this article includes the following figure supplement(s) for figure 4:

**Figure supplement 1.** VR navigation task histology and responses.

**Figure supplement 2.** Trial-by-trial correlation of dopamine slope vs previous inter-reward interval (IRI) in the VR task.

produce weaker ramps with repeated training (*Guru et al., 2020*). These results are generally inconsistent with the stable ramps that we observed in Pavlovian conditioning across eight days (*Figure 1L*). A speculative explanation might be that when the timescales of events vary considerably (e.g. during early experience in instrumental tasks due to variability in action timing), animals use a short eligibility trace time constant to account for the potential non-stationarity of the environment. With repeated exposure, the experienced stationarity of the environment might increase the eligibility trace time constant, thereby complicating its relationship with the ITI. Alternatively, as suggested previously (*Collins et al., 2016*; *Guru et al., 2020*), repeated navigation may result in automated behavior that ignores the progress towards reward, thereby minimizing the calculation of associations of spatial locations with reward.

While the focus of this study was environmental timescales set by the ITI, we also assumed that a dynamic sequence of external cues signaling temporal or spatial proximity to reward would be required for dopamine ramps to occur. Indeed, our results that dopamine ramps occur in the short ITI/dynamic, but not fixed, tone condition corroborate this assumption, as well as results from other Pavlovian conditioning experiments utilizing dynamic cues in a head-fixed setup (*Kim et al., 2020*; *Mikhael et al., 2022*). In contrast, experiments involving freely moving animals do not require explicitly dynamic cues because the sensory feedback from navigating through the environment presumably functions in the same way to indicate proximity to reward. Future experiments can investigate this further by characterizing the role of sensory feedback indicating reward proximity in mediating dopamine ramps. One set of observations superficially inconsistent with our assumption of the necessity of a sequence of external cues is that ramping dopamine dynamics can be observed even when only internal states signal reward proximity (e.g. timing a delayed action; *Guru et al., 2020*; *Hamilos et al., 2021*; *Gao et al., 2021*). In these cases, however, animals were required to actively keep track of the passage of time, which therefore strengthens an internal progression of neural states signaling temporal proximity to reward. We speculate that once learned, these internal states could serve the role of external cues in the ANCCR framework. Previously, we argued against this kind of assumption in learning theories (*Namboodiri, 2022*). Our earlier position was that it is problematic to assume fixed internal states that pre-exist and provide a scaffold for learning, such as in temporal difference learning. This is because these pre-existing states would need to already incorporate information that can only be acquired during the course of learning (*Namboodiri, 2022*). Unlike this position, here we are merely speculating that after learning, an internal progression of states can serve the function of externally signaled events. Similarly, we have previously postulated that such an internal state exists during omission of a predicted reward, but only after learning of the cue-reward association (*Jeong et al., 2022*).

Though the experiments in this study were motivated by the ANCCR framework, they were not conducted to discriminate between theories. As such, it is also possible to rationalize these results in the context of other models of dopamine ramps. In the value model, dopamine is thought to represent the discounted sum of future rewards (*Howe et al., 2013*; *Hamid et al., 2016*; *Mohebi et al., 2019*; *Lloyd and Dayan, 2015*). The shape of this value function, and thus the predicted dopamine dynamics, is determined by the discount factor, $\gamma$. If there is a low $\gamma$ (i.e. greater discounting), then the corresponding value function produces a steeper ramp. Consequently, it is possible to use the value model to explain our results if one assumes that a shorter ITI causes greater temporal discounting. The basis for such an assumption is unclear, though it has been suggested that the overall temporal discounting in an environment depends on reward rates (*Namboodiri et al., 2014*; *Williams*

*et al., 2017*). Furthermore, we do not find evidence for dopamine ramps acting as a value signal to directly increase motivation. This is because we find similar trial-related behaviors in conditions with and without dopamine ramps. Thus, ITI-dependent emergence of dopamine ramps for the same trial parameters provides strong constraints for the motivational role of dopamine ramps.

Substantial efforts have been made to account for the phenomenon of dopamine ramps as a temporal difference RPE (*Kim et al., 2020*; *Lloyd and Dayan, 2015*; *Mikhael et al., 2022*; *Gershman, 2014*; *Masset et al., 2025*). As with the value model, simulated dopamine responses in temporal difference RPE models are also modulated by the discount factor, γ. It has been proposed that temporal discounting in the dopamine system depends on the cue-reward delay (*Sousa et al., 2025*). In our experiment, however, the cue-reward delay is not the key variable determining the presence of ramps; instead, it is the ITI. Another work has also proposed that a spectrum of discount factors can explain diverse activity profiles of single dopamine neurons (*Masset et al., 2025*). Specifically, monotonic upward ramps were simulated using a high γ (i.e. weaker discounting). Therefore, in this model, one would need to assume that shorter ITIs cause weaker temporal discounting to produce steeper ramps. Notably, this is in the opposite direction as the value model. Overall, it is unclear whether any fundamental principle predicts an ITI-dependent change in temporal discounting in the dopamine system to allow RPE to explain our results. Similarly, whether other models of dopamine ramps (*Hamid et al., 2021*; *Lloyd and Dayan, 2015*; *Jakob et al., 2022*) can capture an ITI-dependent emergence of dopamine ramps remains to be explored.

While it is thus possible to rationalize our results using alternative theories of dopamine, ANCCR provides a principled and parsimonious explanation. Given that the foundation of ANCCR is looking back in time for causes of rewards, it is clear that differences in memory maintenance via eligibility traces will have profound implications on predicted dopamine signaling. For example, when the ITI is short and rewards are being frequently delivered, it intuitively makes sense that the eligibility trace time constant needs to be small; this is because the time window over which one would want to search for potentially causal cues is going to be shorter in this situation. We formalize this intuition by postulating that the eligibility trace time constant adapts to the overall event rates for efficient coding (**Appendix 1**). In the case of a dynamic progression of cues, the cues closer in time to the reward will have higher causal power, and thus higher ANCCR, resulting in a dopamine ramp.

Our ANCCR simulations motivated the experiments, but we did not explicitly intend to fit the data. Accordingly, there are several details of the experiments that we did not include in the simulations. First, animals were trained initially using a long ITI (Pavlovian) or medium ITI (VR). This may explain a discrepancy between the simulations and experimental results: the cue onset response in the short ITI condition is small but positive in the experiment but negative in ANCCR. This discrepancy may be because the cue onset was already learned to be meaningful prior to the short ITI condition, thereby resulting in a stronger cue onset response in the experimental data. Further, we did not explicitly model potential trial-by-trial changes in eligibility trace time constant, sensory noise, internal threshold, local mechanisms controlling dopamine release, or sensor dynamics. Thus, we did not expect to capture all experimental observations in the motivating simulations. Regardless of such considerations, the current results provide a clear constraint for dopamine theories and demonstrate that an underappreciated experimental variable determines the emergence of mesolimbic dopamine ramps.

## Methods

**Key resources table**

| Reagent type (species) or resource | Designation | Source or reference | Identifiers | Additional information |
|---|---|---|---|---|
| Strain, strain background (*Mus musculus*, both sexes) | C57BL/6 J | Jackson Laboratory | RRID:IMSR_JAX:000664 | |
| Recombinant DNA reagent | AAVDJ-CAG-dLight1.3b | *Patriarchi et al., 2018* | | |
| Software, algorithm | B-CALM | *Zhou et al., 2024* | RRID:SCR_023884 | |
| Software, algorithm | Doric Neuroscience Studio | Doric Lenses | RRID:SCR_018569 | |
| Software, algorithm | pyPhotometry | *Akam and Walton, 2019* | RRID:SCR_022940 | |

*Continued on next page*

*Continued*

| Reagent type (species) or resource | Designation | Source or reference | Identifiers | Additional information |
|---|---|---|---|---|
| Software, algorithm | BonVision | *Lopes et al., 2021* | RRID:SCR_021534 | |
| Software, algorithm | Python | https://www.python.org/ | RRID:SCR_008394 | |

## Animals

All experimental procedures were approved by the Institutional Animal Care and Use Committee at UCSF and followed guidelines provided by the NIH Guide for the Care and Use of Laboratory Animals. Data from a total of 27 adult wild-type C57BL/6 J mice (#000664, Jackson Laboratory) were included in analysis across experiments: nine mice (four females, five males) were used for the first cohort of Pavlovian conditioning, nine mice (four females, five males) were used for the second cohort of Pavlovian conditioning, and nine mice (six females, three males) were used for the VR task. Data from two additional mice were excluded from analysis: one from the Pavlovian conditioning experiment for failing to display any learning in the form of anticipatory licking and one from the VR task experiment for fiber mistargeting outside of the nucleus accumbens. Following surgery, mice were single-housed in a reverse 12 hr light/dark cycle. Mice received environmental enrichment and had ad libitum access to standard chow. To increase motivation, mice underwent water deprivation. During deprivation, mice were weighed daily and given enough fluids to maintain ~85% of their baseline weight.

## Surgeries

Surgical procedures were always done under aseptic conditions. Induction of anesthesia was achieved with 3% isoflurane, which was maintained at 1–2% throughout the duration of the surgery. Mice received subcutaneous injections of carprofen (5 mg/kg) for analgesia and lidocaine (1 mg/kg) for local anesthesia of the scalp prior to incision. A unilateral injection (Nanoject III, Drummond) of 500 nL of dLight1.3b (*Patriarchi et al., 2018*; AAVDJ-CAG-dLight1.3b, $2.4 \times 10^{13}$ GC/mL diluted 1:10 in sterile saline) was targeted to the NAcC using the following coordinates from bregma: AP 1.3, ML +/-1.4, DV –4.55. The glass injection pipette was held in place for 10 min prior to removal to prevent the backflow of virus. After viral injection, an optic fiber (NA 0.66, 400 µm, Doric Lenses) was implanted 100 µm above the site of injection. Subsequently, a custom head ring for head fixation was secured to the skull using screws and dental cement. Mice recovered and were given at least three weeks before starting behavioral experiments. After completion of experiments, mice underwent transcardial perfusion and subsequent brain fixation in 4% paraformaldehyde. Fiber placement was verified using 50 µm brain sections under a Keyence microscope for subsequent visualization (*Figure 1—figure supplement 2A*, *Figure 2—figure supplement 2A*, *Figure 4—figure supplement 1A*).

## Behavior

All behavioral experiments took place during the dark cycle in dark, soundproof boxes with white noise playing to minimize any external noise. Prior to starting Pavlovian conditioning, water-deprived mice underwent 1–2 days of random rewards training to get acclimated to our head-fixed behavior setup (*Zhou et al., 2024*). In a training session, mice received 100 sucrose rewards (~3 µL, 15% in water) at random time intervals taken from an exponential distribution averaging 12 s. Mice consumed sucrose rewards from a lick spout positioned directly in front of their mouths. This same spout was used for lick detection. After completing random rewards, mice were trained on Pavlovian conditioning. An identical trial structure was used across all conditions, consisting of an auditory tone lasting 8 s followed by a delay of 1 s before sucrose reward delivery. Two variables of interest were manipulated—the length of the ITI (long or short) and the type of auditory tone (fixed or dynamic)—resulting in four conditions: long ITI/fixed tone (LF), long ITI/dynamic tone (LD), short ITI/dynamic tone (SD), and short ITI/fixed tone (SF). In the first cohort (*Figure 1*), mice began with the LF condition (mean 7.4 days, range 7–8) before progressing to the LD condition (mean 6.1 days, range 5–11), the SD condition (8 days), and finally the SF condition (8 days). In the second cohort (*Figure 2*), the experimental order was switched such that mice began with the LF condition before moving on to the SD condition and ending with the LD condition (8 days for each condition). The ITI was defined as the period between reward delivery and the subsequent trial's cue onset. In the long ITI conditions, the ITI was drawn from a truncated

exponential distribution with a mean of 55 s, maximum of 186 s, and minimum of 6 s. The short ITIs were similarly drawn from a truncated exponential distribution, averaging 8 s with a maximum of 12 s and minimum of 6 s. While mice had 100 trials per day in the short ITI conditions, long ITI sessions were capped at 40 trials due to limitations on the amount of time animals could spend in the head-fixed setup. For the fixed tone conditions, mice were randomly divided into groups presented with either a 3 kHz or 12 kHz tone. While the 12 kHz tone played continuously throughout the entire 8 s, the 3 kHz tone was pulsed (200ms on, 200ms off) to make this lower frequency tone more obvious to the mice. For the dynamic tone conditions, the tone frequency either increased (dynamic up↑ starting at 3 kHz) or decreased (dynamic down↓ starting at 12 kHz) by 80 Hz every 200ms, for a total change of 3.2 kHz across 8 s. Mice with the 3 kHz fixed tone had the dynamic up↑ tone, whereas mice with the 12 kHz fixed tone had the dynamic down↓ tone. This dynamic change in frequency across the 8 s was intentionally designed to indicate to the mice the temporal proximity to reward, which is thought to be necessary for ramps to appear in a Pavlovian setting.

For the VR task, water-deprived mice were head-fixed above a low-friction belt treadmill. A magnetic rotary encoder attached to the treadmill was used to measure the running velocity of the mice. In front of the head-fixed treadmill setup, a virtual environment was displayed on a high-resolution monitor (20" screen, 16:9 aspect ratio) using BonVision *Lopes et al., 2021* to look like a dead-end hallway with a patterned floor, walls, and ceiling. The different texture patterns in the virtual environment were yoked to running velocity such that it appeared as though the animal was travelling down the hallway. Upon reaching the end of the hallway, the screen would turn fully black and mice would receive sucrose reward delivery from a lick spout positioned within reach in front of them. The screen remained black for the full duration of the ITI until the reappearance of the starting frame of the virtual hallway signaled the next trial onset. To train mice to engage in this VR task, they began with a 10 cm long virtual hallway. This minimal distance requirement was chosen to make it relatively easy for the mice to build associations between their movement on the treadmill, the corresponding visual pattern movement displayed on the VR monitor, and reward deliveries. Based on their performance throughout training, the distance requirement progressively increased by increments of 5–20 cm across days until reaching a maximum distance of 67 cm. Training lasted an average of 21.4 days (range 11–38 days), ending once mice could consistently run down the full 67 cm virtual hallway for three consecutive days. The ITIs during training ("med ITI") were randomly drawn from a truncated exponential distribution with a mean of 28 s, maximum of 90 s, and minimum of 6 s. Following training, mice were randomly divided into two groups with identical trials but different ITIs (long or short). Again, both ITIs were randomly drawn from truncated exponential distributions: long ITI (mean 62 s, max 186 s, min 6 s) and short ITI (mean 8 s, max 12 s, min 6 s). After 8 days of the first ITI condition, mice switched to the other condition for an additional 8 days. There were 50 trials per day in both the long and short ITI conditions.

## Fiber photometry

Beginning three weeks after viral injection, dLight photometry recordings were performed with either an open-source (PyPhotometry *Akam and Walton, 2019*) or commercial (Doric Lenses) fiber photometry system. Excitation LED light for wavelengths of 470 nm (dopamine-dependent dLight signal) and 405 nm (dopamine-independent isosbestic signal) was sinusoidally modulated via an LED driver and integrated into a fluorescence minicube (Doric Lenses). The same minicube was used to detect incoming fluorescent signals at a 12 kHz sampling frequency before demodulation and downsampling to 120 Hz. Excitation and emission light passed through the same low autofluorescence patchcord (400 μm, 0.57 NA, Doric Lenses). Light intensity at the tip of this patch cord was consistently ~40 μW across days. For Pavlovian conditioning, the photometry software received a TTL signal for the start and stop of the session to align the behavioral and photometry data. For alignment in the VR task, the photometry software received a TTL signal at each reward delivery.

## Data analysis
### Behavior

Licking was the behavioral readout of learning used in Pavlovian conditioning. The lick rate was calculated by binning the number of licks every 100ms. A smoothed version produced by Gaussian filtering is used to visualize lick rate in PSTHs (*Figure 1F*, *Figure 2—figure supplement 3A*, *Figure 4—figure*

*supplement 1D*). Anticipatory lick rate for the last three days combined per condition was calculated by subtracting the average baseline lick rate during the 1 s before cue onset from the average lick rate during the trace period 1 s before reward delivery (*Figure 1G*, *Figure 2—figure supplement 3B*). The same baseline subtraction method was used to calculate the average lick rate during the 3–8 s post-cue onset period (*Figure 2—figure supplement 3C*).

Running velocity, rather than licking, was the primary behavioral readout of learning for the VR task. Velocity was calculated as the change in distance per time. Distance measurements were sampled every 50ms throughout both the trial and ITI periods. Average PSTHs from the last three days per condition were used to visualize velocity aligned to trial onset (*Figure 4D*) and reward delivery (*Figure 4F*). The change in velocity at trial onset was calculated by subtracting the average baseline velocity (baseline being 1 s before trial onset) from the average velocity between 1–2 s after trial onset (*Figure 4E*). Pre-reward velocity was the mean velocity during the 1 s period before reward delivery (*Figure 4G*).

The inter-trial interval (ITI) used throughout is defined as the time period between the previous trial reward delivery and the current trial onset (*Figure 1C*, *Figure 4—figure supplement 1B*). The inter-reward interval (IRI) is defined as the time period between the previous trial reward delivery and the current trial reward delivery (*Figure 4—figure supplement 1B*). For the previous IRI vs trial slope analysis (*Figure 4—figure supplement 2*), IRI outliers were removed from analysis if they were more than three standard deviations away from the mean of the original IRI distribution. Finally, trial durations in the VR task were defined as the time it took for mice to run 67 virtual cm from the start to the end of the virtual hallway (*Figure 4—figure supplement 1B–C*).

## Dopamine

To analyze dLight fiber photometry data, first a least-squares fit was used to scale the 405 nm signal to the 470 nm signal. Then, a percentage dF/F was calculated as follows: dF/F = (470 – fitted 405) / (fitted 405) * 100. This session-wide dF/F was then used for subsequent analysis. The onset peak dF/F (*Figure 1J*, *Figure 4I*) was calculated by finding the maximum dF/F value within 1 s after onset and then subtracting the average dF/F value during the 1 s interval preceding onset (last three days per condition combined). For each trial in Pavlovian conditioning, the time-aligned dLight dF/F signal during the 'ramp window' of 3–8 s after cue onset was fit with linear regression to obtain a per-trial slope. These per-trial slopes were then averaged for each day separately (*Figure 1L*) or for the last three days in each condition (*Figure 1M*) for subsequent statistical analysis. A smoothing Gaussian filter was applied to the group average (*Figure 1I*, *Figure 2B*) and example trial (*Figure 1K*) dLight traces for visualization purposes.

Distance, rather than time, was used to align the dLight dF/F signal in the VR task. Virtual distances were sampled every 30ms, while dF/F values were sampled every 10ms. To sync these signals, the average of every three dF/F values was assigned to the corresponding distance value. Any distance value that did not differ from the previous distance value was dropped from subsequent analysis (as was its mean dF/F value). This was done to avoid issues with averaging if the animal was stationary. For each trial in the VR task, the distance-aligned dLight dF/F signal during the 'ramp window' of 20–57 cm from the start of the virtual hallway was fit with linear regression to obtain a per-trial slope. These per-trial slopes were then averaged for each day separately (*Figure 4L*) or for the last 3 days in each condition (*Figure 4M*) for subsequent statistical analysis. To visualize the group-averaged distance-aligned dLight trace (*Figure 4J*) and example trial traces (*Figure 4K*), the mean dF/F was calculated for every 1 cm after rounding all distance values to the nearest integer.

## Simulations

We previously proposed a learning model called Adjusted Net Contingency of Causal Relation (ANCCR; *Jeong et al., 2022*), which postulates that animals retrospectively search for causes (e.g. cues) when they receive a meaningful event (e.g. reward). ANCCR measures this retrospective association, which we call predecessor representation contingency (PRC), by comparing the strength of memory traces for a cue at rewards ($M_{\leftarrow cr}$; *Equation 1*) to the baseline level of memory traces for the same cue updated continuously ($M_{\leftarrow c-}$; *Equation 2*).

$$M_{\leftarrow cr} \equiv M_{\leftarrow cr} + \alpha \left[ E_{\leftarrow cr} - M_{\leftarrow cr} \right] \tag{1}$$

$$M_{\leftarrow c-} \equiv M_{\leftarrow c-} + \alpha_0[E_{\leftarrow c-} - M_{\leftarrow c-}] \tag{2}$$

$$PRC_{\leftarrow cr} = M_{\leftarrow cr} - M_{\leftarrow c-} \tag{3}$$

$\alpha$ and $\alpha_0$ are learning rates and the baseline samples are updated every $dt$ seconds. $E_{\leftarrow ci}$ represents eligibility trace of cue ($c$) at the time of event $i$ and $E_{\leftarrow c-}$ represents eligibility trace of cue ($c$) at baseline samples updated continuously every $dt$ seconds.

The eligibility trace ($E$) decays exponentially over time depending on decay parameter $T$ (**Equation 4**).

$$E_{\leftarrow i}(t) = \sum_{t_i \leq t} e^{-\left(\frac{t - t_i}{T}\right)} \tag{4}$$

where $t_i \leq t$ denotes the moments of past occurrences of event $i$.

In **Appendix 1**, we derived a simple rule for the setting of $T$ based on event rates. For the tasks considered here, this rule translated to a constant multiplied by IRI. We have shown in a revised version of a previous study (**Burke et al., 2023**) that $\alpha = 1 - (1 - \alpha_0)^{\frac{IRI}{dt}}$ during initial learning.

To mimic the dynamic tone condition, we simulated the occurrence of 8 different cues in a sequence with a 1 s interval between each cue. We used 1 s intervals between cues because real animals are unlikely to detect the small change in frequency occurring every 200ms in the dynamic tone, and we assumed that a frequency change of 400 Hz in 1 s was noticeable to the animals. We included the offset of the last cue as an additional cue. This is based on observation of animal behavior, which showed a sharp rise in anticipatory licking following the offset of the last cue (**Figure 1F–G**). Inter-trial interval was matched to the actual experimental conditions, averaging 2 s for the short dynamic condition and 49 s for the long dynamic condition, with an additional 6 s fixed consummatory period. This resulted in 17 s IRI for short dynamic condition and 64 s IRI for long dynamic condition on average. 1000 trials were simulated for each condition, and the last 100 trials were used for analysis. Following parameters were used for simulation: $w = 0.5$, $b_{cues} = 0$, $b_{reward} = 0.5$, $threshold = 0.2$, $T = 0.2*$IRI, $\alpha_0 = 5 \times 10^{-3}$, $\alpha_R = 1$, dt = 0.2 s.

## Statistics

All statistical tests were run on Python 3.11 using the scipy (version 1.10) package. Full details related to statistical tests are included in **Supplementary file 1**. Data presented in figures with error bars represent mean ± SEM. Significance was determined using 0.05 for α. *p<0.05, **p<0.01, ***p<0.001, ns p>0.05.

## Acknowledgements

We thank J Berke and members of the Namboodiri laboratory for helpful discussions. This project was supported by the NIH (grants R00MH118422 and R01MH129582 to VMKN), Alfred P Sloan Fellowship, Pew Biomedical Scholarship, Klingenstein-Simons Fellowship, the Scott Alan Myers Endowed Professorship (to VMKN), the NSF (graduate research fellowship to JRF), and the UCSF Discovery Fellowship (JRF). The authors have no competing interests.

## Additional information

### Funding

| Funder | Grant reference number | Author |
| --- | --- | --- |
| National Institute of Mental Health | R00MH118422 | Vijay Mohan K Namboodiri |
| National Institute of Mental Health | R01MH129582 | Vijay Mohan K Namboodiri |
| Alfred P Sloan Foundation | Alfred P Sloan Fellowship | Vijay Mohan K Namboodiri |

| Funder | Grant reference number | Author |
|---|---|---|
| Pew Charitable Trusts | Pew Biomedical Scholarship | Vijay Mohan K Namboodiri |
| Klingenstein-Simons Fellowship | | Vijay Mohan K Namboodiri |
| Scott Alan Myers Endowed Professorship | | Vijay Mohan K Namboodiri |
| National Science Foundation | GRFP | Joseph R Floeder |
| University of California, San Francisco | Discovery Fellowship | Joseph R Floeder |

The funders had no role in study design, data collection and interpretation, or the decision to submit the work for publication.

### Author contributions

Joseph R Floeder, Conceptualization, Data curation, Formal analysis, Investigation, Visualization, Methodology, Writing – original draft, Writing – review and editing; Huijeong Jeong, Software, Investigation, Visualization, Writing – review and editing; Ali Mohebi, Resources, Methodology, Writing – review and editing; Vijay Mohan K Namboodiri, Conceptualization, Resources, Supervision, Funding acquisition, Validation, Methodology, Writing – original draft, Project administration, Writing – review and editing

### Author ORCIDs

Joseph R Floeder  https://orcid.org/0000-0002-6800-7730
Huijeong Jeong  https://orcid.org/0000-0003-1219-4191
Ali Mohebi  https://orcid.org/0000-0001-7291-3448
Vijay Mohan K Namboodiri  https://orcid.org/0000-0002-6674-610X

Reviewer #1 (Public review): https://doi.org/10.7554/eLife.98666.3.sa1
Reviewer #2 (Public review): https://doi.org/10.7554/eLife.98666.3.sa2
Reviewer #3 (Public review): https://doi.org/10.7554/eLife.98666.3.sa3
Author response https://doi.org/10.7554/eLife.98666.3.sa4

## Additional files

### Supplementary files

MDAR checklist

Supplementary file 1. Statistical details.

### Data availability

All data for this study are publicly available on the DANDI Archive at https://dandiarchive.org/dandiset/001552. All codes used for analysis and visualization are publicly available on GitHub (https://github.com/namboodirilab/dopamine_ramps; copy archived at *namboodirilab, 2025*).

The following dataset was generated:

| Author(s) | Year | Dataset title | Dataset URL | Database and Identifier |
|---|---|---|---|---|
| Floeder J | 2025 | Floeder et al (2025) Mesolimbic dopamine ramps reflect environmental timescales | https://doi.org/10.48324/dandi.001552/0.250828.1728 | DANDI, 10.48324/dandi.001552/0.250828.1728 |

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

## Appendix 1

### Setting of eligibility trace time constant

It is intuitively clear that the eligibility trace time constant $T$ needs to be set to match the timescales operating in the environment. This is because if the eligibility trace decays too quickly, there will be no memory of past events, and if it decays too slowly, it will take a long time to correctly learn event rates in the environment. Further, the asymptotic value of the baseline memory trace of event $x$, $M_{\leftarrow x-}$ for an event train at a constant rate $\lambda_x$ with average period $t_x$ is $T/t_x = T\lambda_x$. This means that the neural representation of $M_{\leftarrow x-}$ will need to be very high if $T$ is very high and very low if $T$ is very low. Since every known neural encoding scheme is non-linear at its limits with a floor and ceiling effect (e.g. firing rates can't be below zero or be infinitely high), the limited neural resource in the linear regime should be used appropriately for efficient coding. A linear regime of operation for $M_{\leftarrow x-}$ is especially important in ANCCR since the estimation of the successor representation by Bayes' rule depends on the ratio of $M_{\leftarrow x-}$ for different event types. Such a ratio will be highly biased if the neural representation of $M_{\leftarrow x-}$ is in its non-linear range. Assuming without loss of generality that the optimal value of $M_{\leftarrow x-}$ is $M_{opt}$ for efficient linear coding, we can define a simple optimality criterion for the eligibility trace time constant $T$. Specifically, we postulate that the net sum of squared deviations of $M_{\leftarrow x-}$ from $M_{opt}$ for all event types should be minimized at the optimal $T$. The net sum of squared deviations, denoted by $SS$, can be written as

$$SS = \sum_x (M_{\leftarrow x-} - M_{opt})^2 = \sum_x (T\lambda_x - M_{opt})^2 \tag{A1}$$

where the second equality assumes asymptotic values of $M_{\leftarrow x-}$. The minimum of $SS$ with respect to $T$ will occur when $\frac{\rho(SS)}{\varphi T} = 0$. It is easy to show that this means that the optimal $T$ is:

$$T_{opt} = M_{opt} \frac{\sum_x \lambda_x}{\sum_x \lambda_x^2} \tag{A2}$$

For typical cue-reward experiments with each cue predicting reward at 100% probability, $\lambda_{cue} = \lambda_{reward} = 1/IRI$. Substituting into the above equation, we get:

$$T_{opt} = M_{opt}.IRI \tag{A3}$$

Thus, in typical experiments with 100% reward probability, the eligibility trace time constant should be proportional to the IRI or the total trial duration, which is determined by the ITI—the experimental proxy that we manipulate. Please do note, however, that the above relationship is not strictly controlled by the ITI, but by the frequency of repeating events in the environment (i.e., environmental timescale).

Thus, in typical experiments with 100% reward probability, the eligibility trace time constant should be proportional to the IRI or the total trial duration, which is determined by the ITI—the experimental proxy that we manipulate. Please do note, however, that the above relationship is not strictly controlled by the ITI, but by the frequency of repeating events in the environment (i.e., environmental timescale).

