## [Editor Report · eLife Assessment]

Floeder and colleagues provide an **important** investigation that describes the experimental conditions that systematically produce "ramps" in dopamine signaling in the striatum. This somewhat nebulous feature of dopamine has been a significant part of recent theoretical and computational debates attempting to formally describe the different timescales on which dopamine functions. The current results are **convincing** and add context to that ongoing work.

---

## [Referee Report · Reviewer #1 (Public review)]

Summary:

In this study, Floedder et al report that dopamine ramps in both Pavlovian and Instrumental conditions are shaped by reward interval statistics. Dopamine ramps are an interesting phenomenon because at first glance they do not represent the classical reward prediction errors associated with dopamine signaling. Instead, they seem somewhat to bridge the gap between tonic and phasic dopamine, with an intense discussion still being held in the field about what is their actual behavioral role. Here, in tests with head-fixed mice, and dopamine being recorded with a genetically encoded fluorescent sensor in the nucleus accumbens, the authors find that dopamine ramps were only present when intertrial intervals were relatively short and the structure of the task (Pavlovian cue or progression in a VR corridor) contained elements that indicated progression towards the reward (e.g., a dynamic cue). The authors propose that although these findings can be explained by classical theories of dopamine function, they are better explained by their model of Adjusted Net Contingency of Causal Relation (ANCCR). The results of this study provide constraints on future models of dopamine function, and are of high interest to the field.

---

## [Referee Report · Reviewer #2 (Public review)]

In this manuscript by Floeder et al., the authors report a correlation between ITI duration and the strength of a dopamine ramp occurring in the time between a predictive conditioned stimulus and a subsequent reward. They found this relationship occurring within two different tasks with mice, during both a Pavlovian task as well as an instrumental virtual visual navigation task. Additionally, they observed this relationship only in conditions when using a dynamic predictive stimulus. The authors relate this finding to their previously published model ANCCR in which the time constant of the eligibility trace is proportionate to the reward rate within the task.

The relationship between ITI duration and the extent of a dopamine ramp which the authors have reported is very intriguing and certainly provides an important constraint for models for dopamine function. As such, these findings are potentially highly impactful to the field.

---

## [Referee Report · Reviewer #3 (Public review)]

Summary:

Floeder and colleagues measure dopamine signaling in the nucleus accumbens core using fiber photometry of the dLight sensor, in Pavlovian and instrumental tasks in mice. They test some predictions from a recently proposed model (ANCCR) regarding the existence of "ramps" in dopamine that have been seen in some previous research, the characteristics of which remain poorly understood.

They find that cues signaling a progression toward rewards (akin to a countdown) specifically promote ramping dopamine signaling in the nucleus accumbens core, but only when the intertrial interval just experienced was short. This work is discussed in the context of ongoing theoretical conceptions of dopamine's role in learning.

This work is the clearest demonstration to date of concrete training factors that seem to directly impact whether or not dopamine ramps occur. The existence of ramping signals has long been a feature of debates in the dopamine literature and this work adds important context to that. Further, as a practical assessment of the impact of a relatively simple trial structure manipulation on dopamine patterns, this work will be important for guiding future studies. These studies are well done and thoughtfully presented. The additional data, analyses, and discussion in the revised version of the paper add strength and clarity to the conclusions.

The current results raise interesting questions regarding what, if any potential function cue-reward interval dopamine ramps serve. In the current data, licking behavior was similar on different trial types and was not related to ramping activity.

---

## [Author Response]

The following is the authors’ response to the original reviews.

**Reviewer #1 (Public Review):**
Summary:In this study, Floedder et al report that dopamine ramps in both Pavlovian and Instrumental conditions are shaped by reward interval statistics. Dopamine ramps are an interesting phenomenon because at first glance they do not represent the classical reward prediction errors associated with dopamine signaling. Instead, they seem somewhat to bridge the gap between tonic and phasic dopamine, with an intense discussion still being held in the field about what is their actual behavioral role. Here, in tests with head-fixed mice, and dopamine being recorded with a genetically encoded fluorescent sensor in the nucleus accumbens, the authors find that dopamine ramps were only present when intertrial intervals were relatively short and the structure of the task (Pavlovian cue or progression in a VR corridor) contained elements that indicated progression towards the reward (e.g., a dynamic cue). The authors show that these findings are well explained by their previously published model of Adjusted Net Contingency of Causal Relation (ANCCR).Strengths:This descriptive study delineates some fundamental parameters that define dopamine ramps in the studied conditions. The short, objective, and to-the-point format of the manuscript is great and really does a service to potential readers. The authors are very careful with the scope of their conclusions, which is appreciated by this reviewer.

We thank the reviewer for their overall support of the formatting and scope of the manuscript.

Weaknesses:The discussion of the results is very limited to the conceptual framework of the authors' preferred model (which the authors do recognize, but it still is a limitation). The correlation analysis presented in panel l of Figure 3 seems unnecessary at best and could be misleading, as it is really driven by the categorical differences between the two conditions that were grouped for this analysis. There are some key aspects of the data and their relationship with each other, the previous literature, and the methods used to collect them, that could have been better discussed and explored.

We agree with the reviewer that a weakness of the discussion was the limited framing of the results within the ANCCR model. To address this, we have expanded our introduction and discussion sections to provide a more thorough explanation of our model and possible leading alternatives.

We thank the reviewer for pointing out that Figure 3l may be misleading for readers; we removed this panel from the revised Figure 4.

We have further addressed the specific concerns raised by the reviewer in their comments to the authors. Indeed, we agree with the reviewer that the original manuscript was narrow in its focus regarding relationships between different aspects of the data. To more thoroughly explore how key variables – including dopamine ramp slope and onset response as well as licking behavior slope – could relate to each other, we have added Extended Data Figure 8. In this figure, we show that no correlations exist between any of these key variables in either dynamic tone condition; it is our hope that this additional analysis highlights the significance of the clear relationship between dopamine ramp slope and ITI duration.

**Reviewer #2 (Public Review):**
In this manuscript by Floeder et al., the authors report a correlation between ITI duration and the strength of a dopamine ramp occurring in the time between a predictive conditioned stimulus and a subsequent reward. They found this relationship occurring within two different tasks with mice, during both a Pavlovian task as well as an instrumental virtual visual navigation task. Additionally, they observed this relationship only in conditions when using a dynamic predictive stimulus. The authors relate this finding to their previously published model ANCCR in which the time constant of the eligibility trace is proportionate to the reward rate within the task.The relationship between ITI duration and the extent of a dopamine ramp which the authors have reported is very intriguing and certainly provides an important constraint for models for dopamine function. As such, these findings are potentially highly impactful to the field. I do have a few questions for the authors which are written below.

We thank the reviewer for their interest in our findings and belief in their potential to be impactful in the field.

(1) I was surprised to see a lack of counterbalance within the Pavlovian design for the order of the long vs short ITI. Ramping of the lick rate does increase from the long-duration ITIs to the short-duration ITI sessions. Although of course, this increase in ramping of the licking across the two conditions is not necessarily a function of learning, it doesn't lend support to the opposite possibility that the timing of the dynamic CS hasn't reached asymptotic learning by the end of the long-duration ITI. The authors do reference papers in which overtraining tends to result in a reduction of ramping, which would argue against this possibility, yet differential learning of the dynamic CS would presumably be required to observe this effect. Do the authors have any evidence that the effect is not due to heightened learning of the timing of the dynamic CS across the experiment?

We appreciate the reviewer expressing their surprise regarding the lack of counterbalance in our Pavlovian experimental design. We previously did not explicitly do this because the ramps disappeared in the short ITI/fixed tone condition, indicating that their presence is not just a matter of total experience in the task. However, we agree that this is incidental, but not direct evidence. To address this drawback, we repeated the Pavlovian experiment in a new cohort of animals with a revised training order, switching conditions such that the short ITI/dynamic tone (SD) condition preceded the long ITI/dynamic tone (LD) condition (see revised Figure 2a). Despite this change in the training order, the main findings remain consistent: positive dLight slopes (i.e., dopamine ramps) are only observed in the SD condition (Figure 2b-d).

We thank the reviewer for raising these questions regarding licking behavior and learning and their relationship with dopamine ramps. Indeed, a closer look at the average licking behavior reveals subtle differences across conditions (Figure 1f and Extended Data Figure 5a). While the average lick rate during the ramp window does not differ across conditions (Extended Data Figure 5c), the ramping of the lick rate during this window is higher for dynamic tone conditions compared to fixed tone conditions (Extended Data Figure 5d). Despite these differences, we still believe that the main comparison between the dopamine slope in the SD vs LD condition remains valid given their similar lick ramping slopes. Furthermore, our primary measure of learning is not lick slope, but anticipatory lick rate during the 1 s trace preceding reward delivery, which is robustly nonzero across cohorts and conditions (Figure 1g and Extended Data Figure 5b).

Taken together, we hope that the results from our counterbalanced Pavlovian training and more rigorous analysis of lick behavior across conditions provide sufficient evidence to assuage concerns that the differences in ramping dopamine simply reflect differences in learning.

(2) The dopamine response, as measured by dLight, seems to drop after the reward is delivered. This reduction in responding also tends to be observed with electrophysiological recordings of dopamine neurons. It seems possible that during the short ITI sessions, particularly on the shorter ITI duration trials, that dopamine levels may still be reduced from the previous trial at the onset of the CS on the subsequent trial. Perhaps the authors can observe the dynamics of the recovery of the dopamine response following a reward delivery on longer-duration ITIs in order to determine how quickly dopamine is recovering following a reward delivery. Are the trials with very short ITIs occurring within this period that dopamine is recovering from the previous trial? If so, how much of the effect may be due to this effect? It should be noted that the lack of observance of a ramp on the condition of shortduration ITIs with fixed CSs provides a potential control for this effect, yet the extent to which a natural ramp might occur following sucrose deliveries should be investigated.

We thank the reviewer for highlighting the possibility that ramps may be due to the dopamine response recovery following reward delivery. Given that peak reward dopamine responses tend to be larger in long ITI conditions, however, we felt that it was inappropriate to compare post-reward dopamine recovery times across conditions. Instead, we decided to directly compare the dLight slope 2s before cue onset (“pre-cue window,” a proxy for recovery from previous trial) with the dLight slope during our ramp window from 3 to 8s after cue onset (Extended Data Figure 6a). There were no significant differences in pre-cue dLight slope across conditions (Extended Data Figure 6b); this suggests that the ramping slopes seen in the SD condition, but not other conditions, is not simply due to the natural dopamine recovery response following reward delivery. Furthermore, if the dopamine ramps observed in the SD condition were a continuation of the post-reward dopamine recovery from the previous trial, we would expect to see a positive correlation between the dLight slope before and during the cue. However, there is no such correlation between the dLight slopes in the ramp window vs. pre-cue window in the SD condition (Extended Data Figure 6c-d). We believe that this observation, along with the builtin control of the SF condition mentioned by the reviewer, serves as evidence against the possibility of our ramp results being due to a natural ramp after reward delivery.

(3) The authors primarily relate the finding of the correlation between the ITI and the slope of the ramp to their ANCCR model by suggesting that shorter time constants of the eligibility trace will result in more precisely timed predictors of reward across discrete periods of the dynamic cue. Based on this prediction, would the change in slope be more gradual, and perhaps be more correlated with a broader cumulative estimate of reward rate than just a single trial?

To clarify, we do not propose that a smaller eligibility trace time constant results in more precise timing per se. Instead, we believe that the rapid eligibility trace decay from smaller time constants gives greater causal predictive power for later periods in the dynamic cue (see Extended Data Figure 1) since the memory of the earlier periods of the cue is weaker.

We appreciate the reviewer’s curiosity regarding the influence of a broader cumulative estimate of reward vs. only the immediately preceding ITI on dopamine ramp slopes. Indeed, in several instrumental tasks (e.g., Krausz et al., Neuron, 2023), recent reward rate modulates the magnitude of dopamine ramps, making this an important variable to investigate. We chose to use linear regression for each mouse separately to analyze the relationship between the trial dopamine slope and the average previous ITI for the past 1 through 10 most recent trials. In the SD condition, as reported in our earlier manuscript, there was a significantly negative dependence of trial dopamine slope with the single previous ITI (i.e., if the previous ITI was long, the next trial tends to have a weaker ramp). This negative dependence, however, only held for a single previous trial; there was no clear relationship between the per-trial dopamine slope and the average of the past 2 through 10 ITIs (Extended Data Figure 7a). For the LD condition, on the other hand, there is no clear relationship between the per-trial dopamine slope and the average previous ITI for any of the past 1 through 10 trials, with one exception: there is a significantly negative dependence of trial dopamine slope with the average ITI of the previous 2 trials (Extended Data Figure 7b). This longer timescale relationship in the LD condition suggests that the adaptation of the eligibility trace time constant is nuanced and depends on the general ITI length.

In general, though we reason that the eligibility trace time constant should depend on overall event rates, we do not currently propose a real-time update rule for the eligibility trace time constant depending on recent event rates. Accordingly, we are currently agnostic about the actual time scale of history of recent event rate calculation that mediates the eligibility trace time constant. Our experimental results suggest that when the ITI is generally short for Pavlovian conditioning, the eligibility trace time constant adapts to ITI on a rapid timescale. However, only a small fraction of the variability of this rapid fluctuation is captured by recent ITI history. A more thorough investigation of this real-time update rule would need to be done in the future.

**Reviewer #3 (Public Review):**
Summary:Floeder and colleagues measure dopamine signaling in the nucleus accumbens core using fiber photometry of the dLight sensor, in Pavlovian and instrumental tasks in mice. They test some predictions from a recently proposed model (ANCCR) regarding the existence of "ramps" in dopamine that have been seen in some previous research, the characteristics of which remain poorly understood.They find that cues signaling a progression toward rewards (akin to a countdown) specifically promote ramping dopamine signaling in the nucleus accumbens core, but only when the intertrial interval just experienced was short. This work is discussed in the context of ongoing theoretical conceptions of dopamine's role in learning.Strengths:This work is the clearest demonstration to date of concrete training factors that seem to directly impact whether or not dopamine ramps occur. The existence of ramping signals has long been a feature of debates in the dopamine literature and this work adds important context to that. Further, as a practical assessment of the impact of a relatively simple trial structure manipulation on dopamine patterns, this work will be important for guiding future studies. These studies are well done and thoughtfully presented.

We thank the reviewer for recognizing the context that our study adds to the dopamine literature and the potential for our experiments to guide future work.

Weaknesses:It remains somewhat unclear what limits are in place on the extent to which an eligibility trace is reflected in dopamine signals. In the current study, a specific set of ITIs was used, and one wonders if the relative comparison of ITI/history variables ("shorter" or "longer") is a factor in how the dopamine signal emerges, in addition to the explicit length ("short" or "long") of the ITI. Another experimental condition, where variable ITIs were intermingled, could perhaps help clarify some remaining questions.

Though we used ITIs of fixed means, due to the exponential nature of their distribution, we did intermingle ITIs of various durations in both our long and short ITI conditions. The distribution of ITI durations is visualized in Figure 1c for Pavlovian conditioning and Extended Data Figure 9b for VR navigation.

The relative comparison between consecutive ITIs was not something we originally explored, so we thank the reviewer for wondering how it impacts the dopamine signal. To investigate this, we quantified both the change in ITI (+ or - Δ ITI for relatively longer or shorter, respectively) and the change in dopamine ramp slope between consecutive trials in the SD condition (Figure 3d). Across each mouse separately, we found a significantly negative relationship between Δ slope and Δ ITI (Figure 3e-f). Also, the average Δ slope was significantly greater for consecutive trials with a Δ ITI below -1 s compared to trials with a Δ ITI above +1 s (Figure 3g). Altogether, these findings suggest that relative comparison of ITIs does correlate with changes in the dopamine signal; a relatively longer ITI tends to have a weaker ramp, which fits in nicely with the expected inverse relationship between ITI and dopamine ramp slope from our ANCCR model.

In both tasks, cue onset responses are larger, and longer on long ITI trials. One concern is that this larger signal makes seeing a ramp during the cue-reward interval harder, especially with a fluorescence method like photometry. Examining the traces in Figure 1i - in the long, dynamic cue condition the dopamine trace has not returned to baseline at the time of the "ramp" window onset, but the short dynamic trace has. So one wonders if it's possible the overall return to baseline trend in the long dynamic conditions might wash out a ramp.

This is a good point, and we thank the reviewer for raising it. Certainly, the cue onset response is significantly larger in long ITI conditions (see Figure 1i-j and Figure 4h-j). To avoid any bleed over effect, we intentionally chose ramp window periods during later portions of the trial (in line with work from others e.g., Kim et al., Cell, 2020). While the cue onset dopamine pulse seems to have flatlined by the start of the ramp window period, the dopamine levels clearly remain elevated relative to pre-cue baseline. This type of signal has been observed with fiber photometry in other Pavlovian conditioning paradigms with long cue durations (e.g., Jeong et al., Science, 2022). Because of the persistently elevated dopamine levels, it is certainly possible that a ramping signal during the cue is getting washed out; with the bulk fluorescence photometry technique we employed in this study, this possibility is unfortunately difficult to completely rule out. However, the long ITI/fixed tone (LF) condition could serve as a potential control given the overall similarity in the dopamine signal between the LF and LD conditions: both conditions have large cue onset responses with elevated dopamine throughout the duration of the cue (see Extended Data Figures 2c and 3c). Critically, the LD condition lacks a noticeable ramp despite the dynamic tone providing information on temporal proximity to reward, which is thought to be necessary for dopamine ramps to occur. Importantly, regardless of whether a ramp is masked in the long ITI dynamic condition, most studies investigate such a condition in isolation and would report the absence of dopamine ramps. Thus, at a descriptive level, we believe it remains true that observable dopamine ramps are only present when the ITI is short.

Not a weakness of this study, but the current results certainly make one ponder the potential function of cue-reward interval ramps in dopamine (assuming there is a determinable function). In the current data, licking behavior was similar on different trial types, and that is described as specifically not explaining ramp activity.

We agree that this work naturally raises the question of the function of dopamine ramps. However, selective and precise manipulation of only the dopamine ramps without altering other features such as phasic responses, or inducing dopamine dips, is highly technically challenging at this moment; due to this challenge, we intentionally focused on the conditions that determine the presence or absence of dopamine ramps rather than their function. We agree with the reviewer that studying the specific function of dopamine ramps is an interesting future question.

Reviewing Editor:The reviewers felt the results are of considerable and broad interest to the neuroscience community, but that the framing in terms of ANCCR undermined the scope of the findings as did the brief nature of the formatting of the manuscript. In addition, the reviewers felt that the relationship between ramp dynamics, behavior, and ITI conditions requires more in-depth analyses. Relatedly, the lack of counterbalancing of the ITI durations was considered to be a drawback and needs to be addressed as it may affect the baseline. Addressing these issues in a satisfactory manner would improve the assessment of the manuscript to important/convincing.

We truly appreciate the valuable feedback provided on this manuscript by all three reviewers and the reviewing editor. Based on this input, we have significantly revised the manuscript to address the issues brought up by the reviewers. Firstly, we have conducted additional experiments to counterbalance the ITI conditions for Pavlovian conditioning; this strengthened our results by confirming our original findings that ITI duration, rather than training order, is the key variable controlling the presence or absence of dopamine ramps. Secondly, we completed more rigorous analyses to further explore the relationship between dopamine dynamics, animal behavior, and ITI duration; we generally found no significant correlations between these variables, with a notable exception being our main finding between ITI duration and dopamine ramp slope. Finally, we revised and expanded our writing to both explain predictions from our ANCCR model in less technical language and explore how alternative theoretical frameworks could potentially explain our findings. In doing so, we hope that our manuscript is now more accessible and of interest to a broad audience of neuroscience readers.

**Reviewer #1 (Recommendations For The Authors):**
The study could be improved if the authors performed a more detailed comparison of how other theoretical frameworks, beyond ANCCR could account for the observed findings. Also, the correlation analysis presented in the panel I of Figure 3 seems unnecessary and potentially spurious, as the slope of the correlation is clearly mostly driven by the categorical differences between the two ITI conditions, which were combined for the analysis - it's not clear what is the value of this analysis beyond the group comparison presented in the following panel.

Again, we thank the reviewer for elaborating on their concern regarding Figure 3l – we have removed it from the revised Figure 4.

The relationship between ramp dynamics with the behavior and the large differences in cue onset responses between short and long ITI conditions could have been better explored. If I understand correctly the overarching proposal of this and other publications by this group, then the differences in cue responses is determined by the spacing of rewards in a somewhat similar way that the ramps are. So, is there a trial-by-trial correlation between the amplitude of the cue responses and the slope of the ramps? Is there a correlation between any of these two measures with the licking behavior, and if so, does it change with the ITI condition? A more thorough exploration of these relationships would help support the proposal of the primacy of inter-event spacing in determining the different types of dopamine responses in learning.

There are certainly interesting relationships between dopamine dynamics, behavior, and ITI that we failed to explore in our original manuscript – we appreciate the reviewer bringing them up. We found no correlation between dopamine ramp slope and cue onset response in either the SD or LD condition (Extended Data Fig 8a-b). Moreover, we found no correlation between either of these variables and the trial-by-trial licking behavior (Extended Data Fig 8c-f). Finally, there is no relationship between licking behavior and previous ITI duration (Extended Data Fig 8g-h), suggesting that behavioral differences do not account for differences in the dopamine ramp slope. Together, the lack of significant relationships between these other variables highlights the specific, clear relationship between ITI duration and dopamine ramp slope.

Finally, another issue I feel could have been better discussed is how the particular settings of both tasks might be biasing the results. For example, there is an issue to be considered about how the dopamine ramp dynamics reported here, especially the requirement of a dynamic cue for ramps to be present, square with the previous published results by one of the authors - Mohebi et al, Nature, 2019. In that manuscript, rats were executing a bandit task where, to this reviewer's understanding, there was no explicit dynamic cue aside from the standard sensory feedback of the rats moving around in the behavior boxes to approach a nose poke port. Is the idea that this sensory feedback could function as a dynamic cue? If that's the case, then this short-scale, movement-related feedback should also function as a dynamic cue in a freely moving Pavlovian condition, when the animals must also move towards a reward delivery port, right? Therefore, could it be that the experimental "requirement" of a dynamic cue is only present in a head-fixed condition? One could phrase this in a different way to Steelman and potentially further the authors' proposal: perhaps in any slightly more naturalistic setting, the interaction of the animals with their environment always functions as a dynamic cue indicating proximity to reward, and this relationship was experimentally isolated by the use of head fixation (but not explicitly compared with a freely moving condition) in the present study. I think that would be an interesting alternative to consider and discuss, and perhaps explore experimentally at some point.

We thank the reviewer for raising this important point regarding the influence of our experimental settings on our results. At first glance, it could appear that our results demonstrating the necessity of a dynamic cue for ramps in a head-fixed setting do not fit neatly with other results in a freely moving setup (e.g., Collins et al., Scientific Reports, 2016; Mohebi et al., Nature, 2019). Exactly as the reviewer states though, we believe that sensory feedback from the environment in freely moving preparations serves the same function as a dynamic progression of cues. We have considered the implications of methodological differences between head-fixed and freely moving preparations in the discussion section.

**Reviewer #2 (Recommendations For The Authors)*:***
This comment relates indirectly to comment 3, in that the authors intermix theory throughout the manuscript. I think this would be fine if the experiment was framed directly in terms of ANCCR, but the authors specifically mention that this experiment wasn't developed to distinguish between different theories. As such, it seems difficult to assess the scope of the comments regarding theory within the paper because they tend to be specifically related to ANCCR. For instance, the last comment has broad implications of how the ramp might be related to the overall reward rate, an interesting finding that constrains classes of dopamine models rather than evidence just for ANCCR. Perhaps adding a discussion section that allows the authors to focus more on theory would be beneficial for this manuscript.

We appreciate this suggestion by the reviewer. We have updated both our introduction and discussion sections to elaborate more thoroughly on theory.

**Reviewer #3 (Recommendations For The Authors):**
The paper could potentially benefit from the use of more accessible language to describe the conceptual basis of the work, and the predictions, and a bit of reformatting away from the brief structure with lots of supplemental discussion.For example, in the introduction, the line - "Varying the ITI was critical because our theory predicts that the ITI is a variable controlling the eligibility trace time constant, such that a short ITI would produce a small time constant relative to the cue-reward interval (Supplementary Note 1)". As far as I can tell, this is meant to get across the notion that dopamine represents some aspect of the time between rewards - dopamine signals will differ for cues following short vs long intervals between rewards.As written, the language of the paper takes a fair bit of parsing, but the notions are actually pretty simple. This is partly due to the brief format the paper is written in, where familiarity with the previous papers describing ANCCR is assumed.From a readability standpoint, and the potential impact of the paper on a broad audience, perhaps this could be considered as a point for revision.

We thank the reviewer for pointing out the drawbacks of our technical language and brief formatting. To address this, we have removed the majority of the supplementary notes and expanded our introduction and discussion sections. In doing so, we hope that the conceptual foundations of this work, and potential alternative theoretical explanations, are accessible and impactful for a broad audience of readers.